# Double-cuff versus single-cuff bronchial blockers for video-assisted thoracoscopic lobectomy: A randomized controlled trial

Guoliang Tang[1], Xiaobin Li[1], Huaqin Liu[2], Shan Song[3], Fengjiao Zhang[2], Shasha Zhang[2], Chao Zhou [2]*

1 Department of Anaesthesiology, The Nanpi People's Hospital, Cangzhou, Hebei, China, 2 Department of Anaesthesiology, The Fourth Hospital of Hebei Medical University, Shijiazhuang, Hebei, China, 3 Department of Respiratory, The Fourth Hospital of Hebei Medical University, Shijiazhuang, Hebei, China

* zhouchao870607@hebmu.edu.cn

## Abstract

### Background

Bronchial blockers (BBs) manage airways during thoracoscopic lobectomy and isolate lungs by occluding the bronchus at the surgical site. However, intraoperative malpositioning remains a concern. We compared the performance of a new double-cuff BB (DcBB) with an additional cuff near the end of a single-cuff BB (ScBB) with that of ScBBs during thoracoscopic lobectomy.

### Methods

This single-center, randomized, parallel, controlled clinical study enrolled 80 patients undergoing thoracoscopic lobectomy for lung cancer at Nanpi People's Hospital. The patients were randomized into two groups (n = 40 each); DcBB (DcBB used during one-lung ventilation) and ScBB (ScBB used during one-lung ventilation). The primary outcomes were the incidence and number of BB malpositioning. Secondary outcomes time for BB placement and positioning, surgical duration, hypoxemia, incidence of adverse cardiovascular events, grades of lung collapse, surgeon satisfaction, tracheal wall damage assessed during bronchoscopy, contamination rates of the non-surgical lung lobe on the surgical site determined by bronchoscopy, radiographic incidence of pulmonary infiltration on postoperative day 1, and intraoperative blood gas values.

### Results

The incidence of malpositioning was significantly lower in the DcBB than the ScBB group (7.5% *vs*. 30%, P = 0.02). The incidence of malposition was significantly lower in the DcBB cohort (P = 0.002). However, positioning the DcBB required more time

**Data availability statement:** All relevant data are within the manuscript and its Supporting information files.

**Funding:** This study is funded by Health Commission of Hebei Province under grant number 20240190. The funders had no role in study design, data collection and analysis, decision to publish, or preparation of the manuscript.

**Competing interests:** Chao Zhou, Shan Song, Huaqin Liu, Fengjiao Zhang, Shasha Zhang are inventors of the utility model patent for a double-cuff bronchial blocker (Patent name: A Double-Cuff Bronchial Blocker; Patent number: ZL202121177425.1). The double-cuff bronchial blockers (DcBB) used in this study were manufactured by Hangzhou Tappa Medical Technology Co. Ltd. based on the aforementioned patent, and the company provided design modification suggestions for the device, with no financial or material support provided to the authors for this research. All other authors (Guoliang Tang, Xiaobin Li) declare that they have no conflicts of interest. This does not alter our adherence to PLOS ONE policies on sharing data and materials.

$(32.28 \pm 5.37$ *vs*. $19.90 \pm 4.16$ sec, $P < 0.001$). Other secondary outcomes did not differ significantly.

## Conclusions

In patients undergoing thoracoscopic lobectomy, the DcBB was associated with a lower incidence of malposition than the ScBB. The DcBB did not increase tracheal wall injury or cause significant adverse hemodynamic effects, and therefore may serve as a promising optional device for airway management during thoracoscopic lobectomy, particularly in clinical scenarios requiring high placement stability.

## Trial registration

the Chinese Clinical Trial Registry ChiCTR2400086472

---

## Introduction

Lung cancer is one of the most prevalent diseases worldwide [1], and surgical resection is the primary treatment during the early stage [2]. Bronchial blockers (BBs) are airway management tools that are used to isolate lungs during thoracoscopic lobectomy [3]. Effective airway management is achieved by occluding the bronchus on the surgical side. The use of BBs has increased in clinical practice as they cause minimal lung damage and are easy to operate. However, BBs are prone to malpositioning during surgery compared with double-lumen tubes (DLTs) [4]. Therefore, fiberoptic bronchoscopy by anesthesiologists may be required, contributing to interruptions during surgery [5]. Moreover, neither DLT nor BB can effectively protect the non-operative lung or lobe(s) from contamination by the operative lung [6]. The double-cuff BB (DcBB) is a new type of BB with another cuff near the end of a single-cuff BB (ScBB). The two cuffs are positioned in the main bronchus of the surgical lung and the bronchus intermedius of the right lung or the inferior lobar bronchus of the left lung. This increases the contact area between the DcBB and the tracheal wall while isolating the surgical and non-surgical lung lobes on the procedural side. Therefore, we hypothesized that the incidence of DcBB malpositions during thoracoscopic lung surgery would be relatively low and that the DcBB would be able to isolate the non-operative lobe(s) of the operative lung. The performance of this device has not been evaluated in prospective randomized clinical trials as far as we can ascertain. Therefore, we aimed to determine whether the DcBB is more effective than the ScBB during thoracoscopic lobectomy.

## Methods

### Ethics

The Medical Ethics Management Committee of Nanpi People's Hospital approved this trial (ID: NY-LL-2024–001; June 12, 2024), which complied with the ethical principles enshrined in the Declaration of Helsinki (2013 amendment). All included patients

provided written informed consent to participate. We registered our study with the Chinese Clinical Trial Registry on 02/07/2024 before patient enrollment (http://www.chictr.org.cn; registration number: ChiCTR2400086472; principal investigator: Tang Guoliang). The trial proceeded between July 3 and November 15, 2024. The first author was responsible for recruiting participants scheduled for thoracoscopic lobectomy at the hospital. The manuscript was prepared according to the Consolidated Standards of Reporting Trials (CONSORT) statement. The CONSORT flow diagram is presented in Fig 1. All BBs used herein were manufactured by the Hangzhou Tappa Medical Technology Co. Ltd. (Hangzhou, China).

## Participants

We recruited patients with lung cancer who were scheduled to undergo thoracoscopic lobectomy. The inclusion criteria were: American Society of Anesthesiologists (ASA) classifications I–II; age 18–65 years and body mass index (BMI), 18.5–25 kg/m²; no significant abnormalities in lung function, ratio of forced expiratory volume in 1 second to forced vital capacity (FEV1/FVC%) >70%, with either restrictive or obstructive ventilatory dysfunction; no significant abnormalities in cardiac function or cardiovascular disease, ejection fraction >50%; no preoperative anemia or other hematological diseases; no history of radiotherapy or chemotherapy; and provision of written informed consent to participate. The exclusion

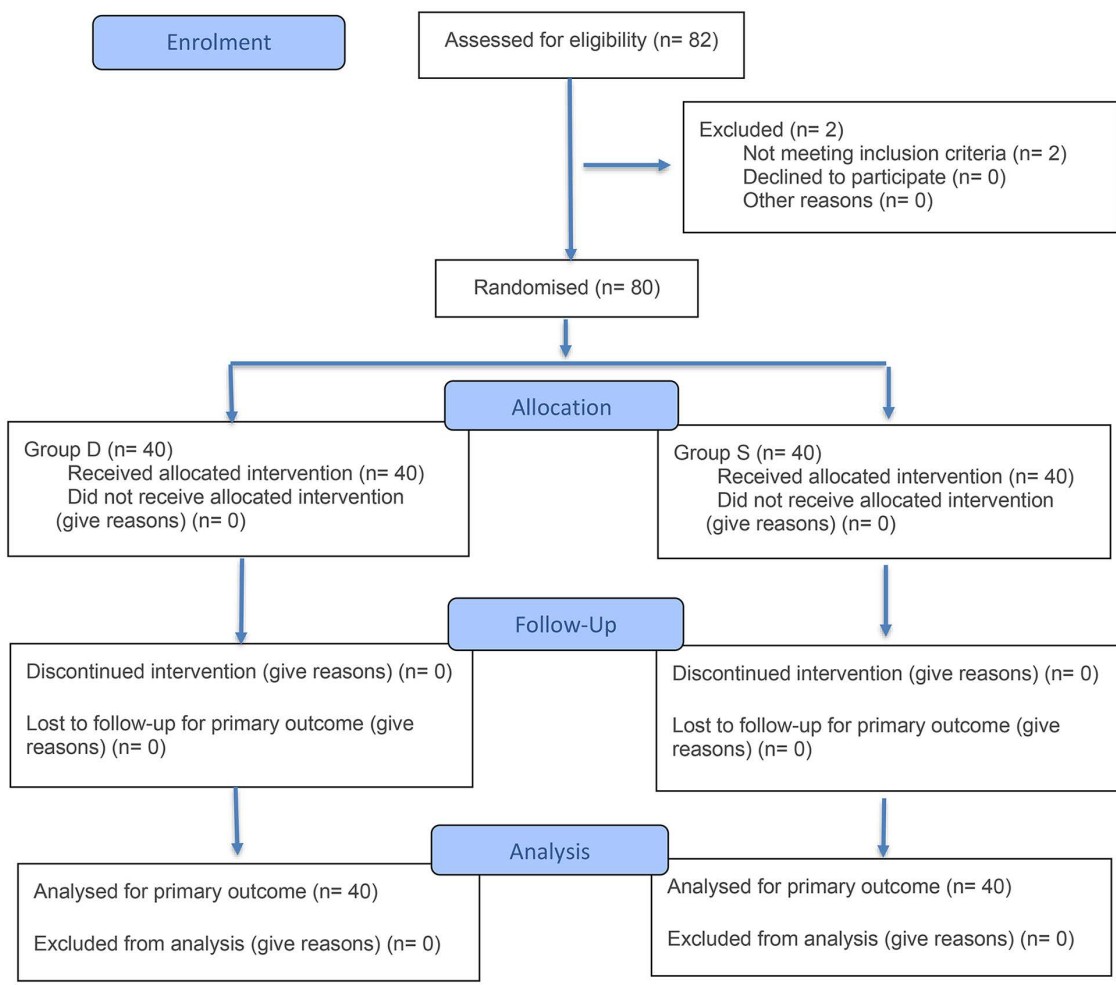

**Fig 1. CONSORT flow diagram.**

criteria were: moderate or severe abnormalities in cardiopulmonary function; history of bronchial asthma and airway hyperreactivity; pulmonary infection, bronchopleural fistula, emphysema, or pulmonary bullae; right main bronchus length <0.5 cm; and tumors or other contraindications to using occlusive devices in the airway.

After enrollment into the trial, all patients were randomly divided into two groups (DcBB group and ScBB group) at a 1:1 allocation ratio, with 40 patients in each group. The randomization sequence was generated by a statistician who was not involved in patient enrollment or data collection, with the help of IBM SPSS Statistics 23.0 for Windows (IBM Corporation, Armonk, NY, USA). Allocation concealment was implemented using sequentially numbered, opaque, and sealed envelopes. Each envelope was made of light-impermeable kraft paper, sealed with glue and stamped with the study-specific seal, and only labelled with a consecutive number (1–80) without any group-related information. Each envelope contained a card indicating the group assignment (DcBB group or ScBB group), and the envelopes were opened only after the patient had been enrolled and baseline data collected by an independent research assistant not involved in the randomization process. Owing to the distinct physical structure differences between the DcBB (double cuffs) and ScBB (single cuff), blinding of the anesthesiologists performing BB placement and the surgeons conducting thoracoscopic lobectomy was not feasible.

## Anesthesia

All patients fasted for 8 h and refrained from drinking fluids for 2 h preoperatively. An operating room nurse established upper limb venous access, and the anesthesiologist connected an IntelliVue MP50 monitor (Royal Dutch Philips Electronics Ltd., Amsterdam, The Netherlands) to record pulse, blood oxygen saturation, and electrocardiograms. Radial artery catheterization proceeded under local anesthesia to monitor arterial blood pressure. Atropine (0.5 mg) was administered intravenously 5 min before inducing anesthesia.

Oxygen was inhaled and nitrogen was exhaled through a mask before inducing anesthesia. The patients were intravenously injected with 0.2–0.4 μg/kg sufentanil, followed by 0.2–0.3 mg/kg etomidate and 0.2 mg/kg cis-atracurium. Mask-assisted ventilation was provided for 3 min after patients lost consciousness.

The airway management plan for the DcBB group was as follows. An 8.0# tracheal tube was orally inserted under direct vision with the cuff placed 2 cm below the vocal cords, then a DcBB was inserted using a fiberoptic bronchoscope at the appropriate site (main bronchus of surgical lung and bronchus intermedius of right lung or inferior lobar bronchus of left lung) in the surgical lung, and the tracheal tube and DcBB were secured (Fig 2).

The airway management plan for the ScBB group was as follows. An 8.0# tracheal tube was inserted orally under direct vision, with the cuff placed 2 cm below the vocal cords, then a 9-Fr tube was placed into the main bronchus of the surgical side through the tracheal tube. A fiberoptic bronchoscope was used to ensure the appropriate position of the ScBB before securing it.

Anesthesia maintenance and recovery procedures were the same for all patients. Anesthesia was maintained in all patients with a continuous infusion of remifentanil (0.05–0.5 μg/kg/min) combined with sevoflurane (1%–3%) inhalation. A bispectral index of 40–60 was maintained, and 0.05 mg/kg cis-atracurium was administered every 30 min. The respiratory parameters during double-lung ventilation comprised tidal volume (Vt) 6–8 mL/kg, respiratory rate (f) 12 breaths/min, and inspiratory-expiratory ratio (I:E) 1:2. The fresh gas flow was 2 L/min with inhaled 100% oxygen ($FiO_2$) to maintain an end-tidal $CO_2$ partial pressure ($PetCO_2$) between 35 and 45 mmHg.

The BBs in laterally positioned patients were examined using a bronchoscope. Lung collapse was induced using a disconnection technique after releasing the pleura [7,8]. The non-surgical side was manually re-expanded, then single-lung ventilation was initiated. The cuff pressure of the BB in both groups was 30 cm $H_2O$. The respiratory parameters during single-lung ventilation were: Vt 4–6 mL/kg; f 15 breaths/min; I:E 1:2; and positive end-expiratory pressure 5 cm $H_2O$. The fresh gas flow rate was 2 L/min, with an $FiO_2$ of 80%, maintaining a $PetCO_2$ between 35–45 mm Hg. After recovery from double-lung ventilation, the BB was removed and tracheal wall damage and rates of contamination (defined as fresh blood

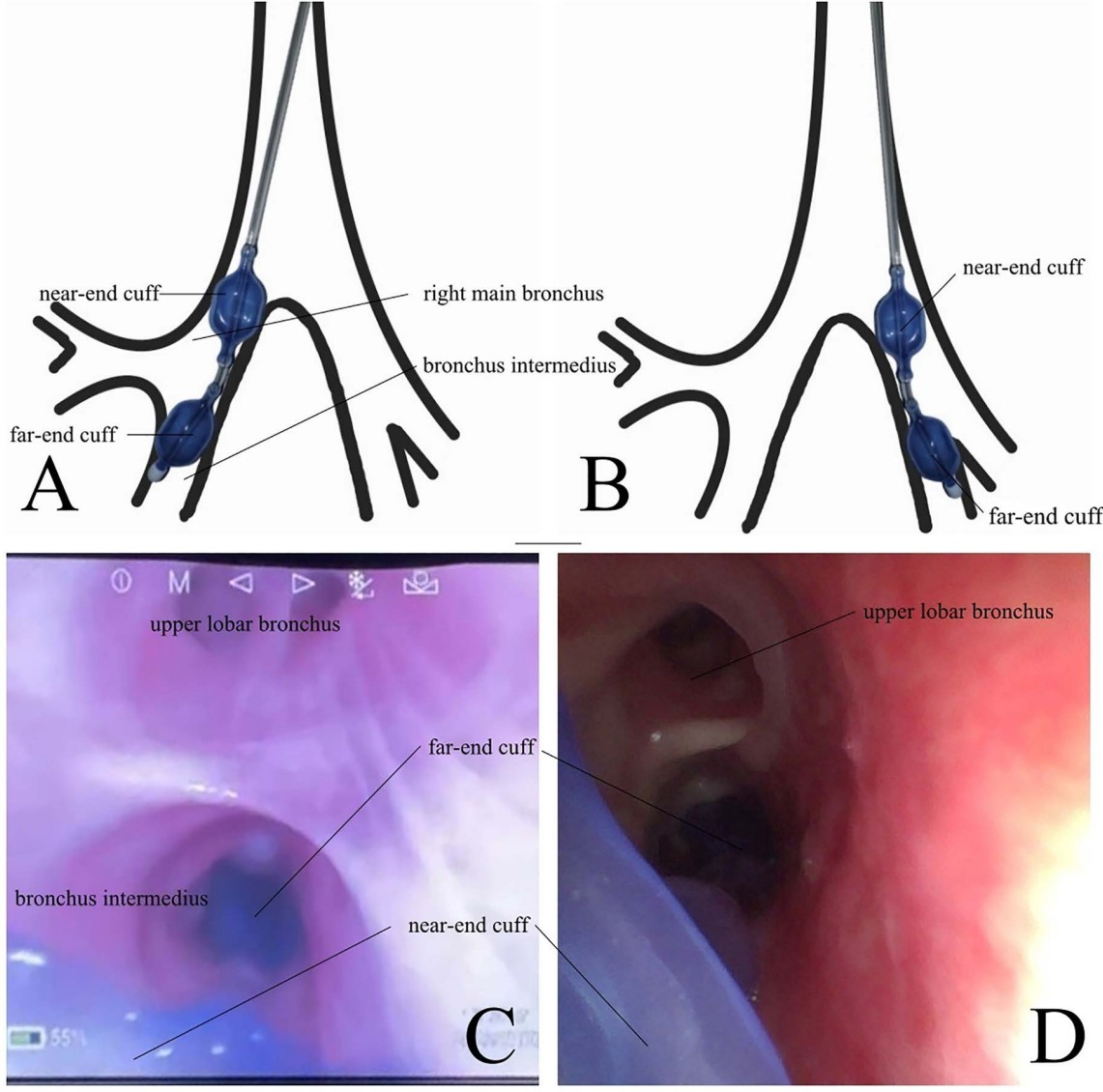

**Fig 2. Position of double-cuff bronchial blocker (DcBB). A:** Far- and near-end cuffs are placed in the bronchus intermedius of the right lung and through a tube in the right main bronchus, respectively, for right lung surgery. **B:** For left lung surgery, the far- and near-end cuffs are respectively placed in the inferior lobar bronchus of left lung and through a tube in the left main bronchus. Bronchoscopic views of the DcBB during right **(C)** and left **(D)** lung surgery.

evidenced during bronchoscopy) of the non-surgical lung lobe on the surgical side were recorded. The endotracheal tube was removed after spontaneous breathing recovery, and patients were returned to the ward when they achieved a Steward score of 6.

## Outcomes

The primary outcomes were the incidence and number of BB malposition. Intraoperatively, a malpositioned BB was considered when the airway pressure suddenly increased, ventilation could not be maintained, or the surgical lung suddenly

inflated [9]. The anesthesiologist confirmed malpositioning using the bronchoscope and repositioned the BB. All cases of suspected or confirmed BB malposition were documented with bronchoscopic images during the operation. Postoperatively, all images were independently reviewed and verified by a second senior anesthesiologist who was not involved in the anesthesia procedure and was blinded to group assignment. Secondary outcomes included time required for BB positioning, surgical duration, hypoxemia, incidence of cardiovascular adverse events, lung collapse grade, surgeon satisfaction, tracheal wall damage assessed by bronchoscopy, contamination rate of the non-surgical lung lobe on the surgical side assessed by bronchoscopy, incidence of pulmonary infiltration on postoperative day (POD) 1 determined by radiography, and intraoperative blood gas results. All patients underwent blood gas analysis after entering the operating room (T0), and after 30 (T1) and 60 (T2) min of single-lung ventilation. Blood oxygen saturation <90% was defined as hypoxemia, and a heart rate <50 or >100 beats per minute during BB placement and blood pressure fluctuations >30% of baseline were defined as adverse cardiovascular events. Lung collapse was graded as excellent, fair, or bad using a proposed classification system [10]. Excellent referred to complete collapse of the lung on the surgical side, satisfactory exposure of the surgical field, and no need for surgical intervention. Fair referred to basic collapse of the lung on the surgical side, some residual gas in the lung, but no ventilation, and relatively satisfactory exposure of the surgical field. Bad referred to no or partial collapse of the lung on the surgical side, affecting the surgical procedure and requiring surgical intervention. Postoperative satisfaction with respect to the operating space was evaluated by the surgeon as Levels I (large operating space, uneventful surgery), II (sufficient operating space to complete the procedure), and III (small operating space that impeded surgery completion without adjustment) [6].

## Analysis

The reported malpositioning rate of BBs during surgery is 25%–75% [10–13]. Our pre-tests indicated a DcBB malpositioning rate of 12%. Therefore, we assumed that the malposition rates in the ScBB and DcBB groups were 40% and 12%, respectively. This assumption is supported by published evidence: a meta-analysis reported a pooled malposition rate of 31.9% for BBs [13], and the dislocation rate of airway devices can exceed 44.6% after lateral positioning [14]. Furthermore, recent trial designs have adopted a 40% malposition rate for BBs in sample size calculation [15]. The formula for comparing the ratios of two independent samples was:

$$n = 0.5 \times \left[ \frac{(u_\alpha + u_\beta)}{\sin^{-1}\sqrt{p_1} - \sin^{-1}\sqrt{p_2}} \right]^2$$

Therefore, each group required 36 participants (power, 0.8; type I error, 0.05; $u_\alpha = 1.96$; $u_\beta = 0.842$). To address the exclusion and loss issues, we increased the number of patients in each group to 40.

Malposition rates (%) were compared using Fisher exact tests. Time required for BB placement and positioning and $PaO_2$ and $PaCO_2$ values were compared between the groups using t-tests, and the results are expressed as means ± standard deviation (SD). The incidences of grades of lung collapse and surgeon satisfaction were compared between the groups using Fisher exact tests. The incidences of hypoxemia, adverse cardiovascular events, tracheal wall injury assessed during bronchoscopy, non-surgical lung lobe contamination on the surgical side determined by bronchoscopy, and pulmonary infiltration on POD 1 were compared between the groups using chi-square tests. Instances of BB malpositioning were compared between the groups using Mann–Whitney U tests, and the results are presented as medians with interquartile ranges (IQRs). Continuous variables with repeated measurements ($PaO_2$ and $PaCO_2$) at three time points were analyzed using the generalized estimating equation to account for within-patient correlation. An independent working correlation matrix was used, and results are reported as mean differences with 95% confidence intervals (CI) and P-values. Values with $P < 0.05$ were considered statistically significant. All data were analyzed using IBM SPSS Statistics 23.0 for Windows.

Intention-to-treat analysis was applied in this study. All patients who underwent randomization were included in the final statistical analysis according to their originally assigned group, with no patients excluded from the analysis set.

Clinical implementation, data collection, and outcome assessment were performed independently by investigators with no involvement in the DcBB device invention. The inventor contributed only to study design and manuscript preparation and did not participate in patient enrollment, intraoperative management, data collection, outcome adjudication, or statistical analysis.

## Results

A total of 82 patients with lung cancer scheduled for video-assisted thoracoscopic lobectomy were screened for this study. Among these, two were excluded for failing to meet the inclusion criteria: one with abnormal pulmonary function and one with a short right main bronchus. Ultimately, 80 patients were enrolled and allocated to the DcBB group (n=40) and ScBB group (n=40) per the randomization protocol. All enrolled patients completed the surgery and postoperative follow-up, with no loss to follow-up between July and September 2024 (Table 1).

### Primary outcomes

Table 2 shows a significantly lower incidence and number of malpositioned cuffs in the DcBB group than in the ScBB group (P=0.02 and P=0.002, respectively; Table 2).

### Secondary outcomes

The duration of BB positioning was significantly longer in the DcBB group than in the ScBB group (P<0.001; Table 3). Between-group differences were not significant. The duration of BB placement, surgical duration, incidence of hypoxemia,

**Table 1. Patient characteristics.**

|  | DcBB group (n=40) | ScBB group (n=40) |
|---|---|---|
| Age, years | 57.45±6.06 | 57.10±6.36 |
| Sex (male/female) | 14/26 | 16/24 |
| Height, cm | 165.78±7.47 | 165.93±9.01 |
| Weight, kg | 63.15±8.03 | 63.08±9.20 |
| ASA physical status |  |  |
| I | 10 | 8 |
| II | 30 | 32 |
| Operative side (left/right) | 16/24 | 18/22 |

ASA, American Society of Anesthesiologists.

**Table 2. Number and incidence of malpositions.**

| Bronchial blocker malpositioning | DcBB group (n=40) | ScBB group (n=40) | Difference (95% CI) | Odds ratio (95% CI) | P |
|---|---|---|---|---|---|
| Number, n (%) | 3 (7.5%) | 12 (30%) | −22.50% (−38.69%, −5.32%) | 0.19 (0.05, 0.68) | 0.01 |
| Instances | 0 (0,0) | 0 (0,2) |  |  | <0.00 |

Results are presented as numbers with ratios (%) or as medians with interquartile ranges. CI, confidence intervals.

**Table 3. Secondary outcomes of groups.**

| | DcBB group (n=40) | ScBB group (n=40) | Difference (95% CI) | Odds ratio (95% CI) | P |
|---|---|---|---|---|---|
| Duration of placing bronchial blocker(s) | 21.70±3.16 | 21.65±3.24 | 0.05 (−1.37, 0.47 | – | 0.94 |
| Duration of positioning bronchial blocker(s) | 32.28±5.37 | 19.90±4.16 | 12.38 (10.24, 14.51) | – | < 0.00 |
| Surgical duration (min) | 136.60±14.85 | 142.68±16.43 | −6.18 (−13.15, 0.80) | – | 0.08 |
| Hypoxemia | 0 | 0 | – | – | – |
| Incidence of adverse cardiovascular events | 0 | 0 | – | – | – |
| Lung collapse grade | | | | | |
| Excellent | 38 | 39 | – | – | 1.00 |
| Fair | 2 | 1 | | | |
| Surgeons' satisfaction | | | | | |
| 1 | 38 | 37 | – | – | 0.61 |
| 2 | 2 | 3 | | | |
| Tracheal wall damage* | 3 | 2 | 0.03 (−0.10, 0.15) | 1.54 (0.24, 9.75) | 1.00 |
| Contamination rate of non-surgical lung lobe on surgical side* | 0 | 1 | 0.05 (−0.07, 0.13) | Not estimated | 1.00 |
| Incidence of pulmonary infiltration on POD 1 | 13 | 14 | 0.03 (−0.18, 0.22) | 0.90 (0.37, 2.20) | 1.00 |

*Determined by bronchoscopy. CI, confidence interval, POD, postoperative day.

cardiovascular adverse events, lung collapse grades, surgeon satisfaction, incidence of tracheal wall damage assessed during bronchoscopy, contamination rate of non-surgical lung lobes on the surgical side determined by bronchoscopy, or the incidence of pulmonary infiltration on POD 1 did not significantly differ between the groups (Table 3). The $PaO_2$ and $PaCO_2$ also did not significantly differ at various time points between the groups (Table 4).

## Discussion

Our study found that the malposition rate of the DcBB was significantly lower than that of the ScBB, and the number of malposition events was also markedly reduced. The underlying mechanism may be related to the unique double-cuff design. Similar to a DLT, the DcBB has two cuffs that provide dual-point fixation. For DLTs, the two cuffs are positioned in the trachea and main bronchus, whereas for the DcBB, the proximal and distal cuffs are placed in the main bronchus and bronchus intermedius (right side) or inferior lobar bronchus (left side), respectively. In both devices, the dual cuffs maintain stable contact with the airway wall, which explains why the DcBB

**Table 4. Blood gas analysis.**

| Variable | Group (n=40 each) | T0 | T1 | T2 |
|---|---|---|---|---|
| $PaO_2$ (mm Hg) | DcBB | 77.65±3.71 | 225.73±46.42 | 246.38±45.96 |
| | ScBB | 76.93±3.87 | 213.58±38.84 | 233.35±40.47 |
| | Mean difference (95% CI) | — | 0.17 (−0.69, 0.85) | 0.02 (−0.71, 0.96) |
| | P-value | | 0.83 | 0.78 |
| $PaCO_2$ (mm Hg) | DcBB | 39.83±1.50 | 41.15±2.14 | 41.20±2.01 |
| | ScBB | 39.85±1.67 | 40.98±2.12 | 41.18±2.46 |
| | Mean difference (95% CI) | — | 0.17 (−0.69, 0.85) | 0.02 (−0.71, 0.96) |
| | P-value | | 0.83 | 0.78 |

$PaCO_2$, partial pressure of arterial carbon dioxide; $PaO_2$, partial pressure of arterial oxygen.

is also resistant to malposition as the DLT [5]. Previous studies have confirmed that the malposition rate of DLTs is lower than that of conventional BBs [16]. A meta-analysis by Kumar et al. further reported a malposition rate of 17.0% for DLTs, which was significantly lower than the 26.4% reported for the EZ blocker [5]. These findings indicate that a double-cuff structure reduces the risk of malposition. In our study, the lower malposition rate with the DcBB may have reduced the frequency of intraoperative interruptions; however, there was no significant between-group difference in operative duration, which may be attributed to the relatively small sample size of this study. We also found that significantly more time was required to position the DcBB than the ScBB. This is because the positions of the two cuffs should be determined when using a DcBB. In particular, when positioning the far-end cuff, the bronchoscope must be maneuvered around the cuff at the near-end of the BB; this confers higher demands on anesthesiologists in terms of bronchoscopic skills [5,16] and contributes to the need for more time when positioning a DcBB. This might decrease as proficiency in using the DcBB increases [5,17,18]. We found that a DcBB can be appropriately placed by positioning only the near-end cuff. However, this may be inconsistent with the original design intent of the DcBB, which was to isolate the surgical from the non-surgical lung lobe on the surgical side [19,20]. We did not find any differences in contamination rates of non-surgical lung lobes on the surgical side between the groups. Therefore, further investigation is needed to determine whether positioning the proximal balloon of the DcBB can reduce the duration of the positioning without increasing the risk of contaminating the non-surgical lung lobe on the surgical side.

We found that both the DcBB and the ScBB achieved good lung collapse effects, consistent with previous findings [21,22]. Moreover, none of the patients developed hypoxemia or cardiovascular adverse events, indicating that DcBBs retain the low malposition rate of DLTs without losing the minimal hemodynamic impact of BBs [17,23,24]. The blood gas analysis results for both groups were within normal ranges. Tracheal damage did not differ significantly between groups, suggesting that DcBBs preserve the minimal lung damage associated with BB use [25]. Regardless of type, the BB should be retracted into the tracheal tube before tracheal incision to avoid cutting the BB [26–28].

This study has some limitations. The single-center design required ensuring consistency among the participants. Thus, we included only patients undergoing lobectomy. Therefore, the applicability of our findings to patients requiring single-lung ventilation should be further explored. The size of the sample was calculated based on the BB malpositioning rate, which resulted in it being relatively small. Blinding of anesthesiologists and surgeons was not feasible in this trial due to the substantial structural differences between the DcBBs and the ScBBs, as the devices could be easily identified by the operators, making concealment of group assignment impossible.

## Conclusion

In patients undergoing thoracoscopic lobectomy, the DcBB was associated with a lower incidence of malposition than the ScBB. The DcBB did not increase tracheal wall injury or cause significant adverse hemodynamic effects, and therefore may serve as a promising optional device for airway management during thoracoscopic lobectomy, particularly in clinical scenarios requiring high placement stability.

## Supporting information

**S1 File. Research protocol.**
(DOCX)

**S2 File. CONSORT 2010 Checklist.**
(DOC)

**S3 File. Data.**
(XLSX)

## Acknowledgments

The authors are grateful to Hangzhou Tappa Medical Technology Co. Ltd. for manufacturing the double-cuff BBs used herein and for proposing helpful modifications to the design.

## Author contributions

**Conceptualization:** Guoliang Tang, Xiaobin Li, Huaqin Liu, Shan Song, Fengjiao Zhang, Shasha Zhang, Chao Zhou.

**Investigation:** Shan Song, Fengjiao Zhang.

**Writing – original draft:** Guoliang Tang, Xiaobin Li, Chao Zhou.

**Writing – review & editing:** Huaqin Liu, Shasha Zhang.

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
