## [Decision Letter · Decision Letter 0]

9 Mar 2026

PONE-D-26-07168Double-cuff versus single-cuff bronchial blockers for video-assisted thoracoscopic lobectomy: a randomised controlled trialPLOS One

Dear Dr. Zhou,

Thank you for submitting your manuscript to PLOS ONE. After careful consideration, we feel that it has merit but does not fully meet PLOS ONE’s publication criteria as it currently stands. Therefore, we invite you to submit a revised version of the manuscript that addresses the points raised during the review process.

We look forward to receiving your revised manuscript.

Kind regards,

Luca Bertolaccini, M.D., Ph.D.

Academic Editor

PLOS One

**Journal Requirements:**

4. We note that you have a patent relating to material pertinent to this article. Please provide an amended statement of Competing Interests to declare this patent (with details including name and number), along with any other relevant declarations relating to employment, consultancy, patents, products in development or modified products etc. Please confirm that this does not alter your adherence to all PLOS ONE policies on sharing data and materials, as detailed online in our guide for authors http://journals.plos.org/plosone/s/competing-interests by including the following statement: "This does not alter our adherence to  PLOS ONE policies on sharing data and materials.” If there are restrictions on sharing of data and/or materials, please state these. Please note that we cannot proceed with consideration of your article until this information has been declared.

6. Thank you for providing your underlying data as Supporting Information.

We note that the data set contains text or data that is not in English. Please note that PLOS is an English-language publisher, so we require data sets to be provided in English as well. Please upload an English-language version of your data set.

This will also allow us to determine if your data follows PLOS standards per our Data Availability policy here: https://journals.plos.org/plosone/s/data-availability

Reviewers' comments:

Reviewer's Responses to Questions

**Comments to the Author**

1. Is the manuscript technically sound, and do the data support the conclusions?

Reviewer #1: Yes

Reviewer #2: Yes

Reviewer #3: Partly

2. Has the statistical analysis been performed appropriately and rigorously? 

Reviewer #1: No

Reviewer #2: Yes

Reviewer #3: No

3. Have the authors made all data underlying the findings in their manuscript fully available?

Reviewer #1: Yes

Reviewer #2: No

Reviewer #3: No

4. Is the manuscript presented in an intelligible fashion and written in standard English?

Reviewer #1: No

Reviewer #2: No

Reviewer #3: Yes

5. Review Comments to the Author

Reviewer #1: This randomized controlled trial compares the performance of a novel double-cuff bronchial blocker (DcBB) with a conventional single-cuff bronchial blocker (ScBB) during video-assisted thoracoscopic lobectomy. The authors demonstrate that the DcBB significantly reduces the incidence of malpositioning without increasing adverse events, highlighting its potential as an effective airway management tool. While the study addresses an important clinical issue and provides valuable insights, several methodological, statistical, and language issues need to be addressed before the manuscript can be considered for publication. My specific comments are as follows.

1. Randomization and Allocation Concealment

The use of a computer-generated random number table to randomly assign patients is documented. However, there is no description of how the randomization sequence was concealed until the time of assignment (e.g., whether opaque, sealed envelopes with sequential numbers were used, or whether a central registration system was employed). This is an important item in the CONSORT statement. Please specify.

2. Statistical Analysis

2-1. Lines 114-117: Because the number of patients who experienced these events was low, Fisher's exact test should be used instead of the chi-square test.

2-2. Table 2: In the "Difference" column, please specify the point estimate in addition to the 95% CI. Furthermore, in the "Risk ratio" column, the calculated values seem to represent the Odds Ratio rather than the Risk Ratio. Please clarify and correct this.

2-3. Table 3: In the "Risk ratio" column, the calculation seems to be based on the number of patients who did NOT experience the events. Please correct this to reflect the occurrence of the events.

3. Language and Formatting Issues

The manuscript requires thorough language editing. Here are specific examples of errors that need to be corrected:

3-1 Abstract (Methods): The sentence "we enrolled and randomized 80 patients..." begins with a lowercase "w".

3-2 Abstract (Methods): There is a missing period before "The primary outcomes": "...(ScBB used during one-lung ventilation) The primary outcomes...".

3-3 Abstract (Results): There is an unnecessary comma in: "...significantly lower in the DcBB, than the ScBB group...".

3-4 Abstract (Results): There is a missing period before "However": "...in the DcBB cohort (P=0.002) However, positioning the DcBB...".

3-5 Line 131: There is a missing period between sentences: "Between-group differences were not significant The duration of BB placement...".

3-6 Lines 139-150: The text contains a major structural issue that leaves sentences disjointed and garbled. For example, a sentence cuts off with "The malposition", continues awkwardly on the next line, and later abruptly inserts fragments like "rate of the DLT is lower than that of BBs¹s and a meta-analysis also found a rate of 17.0%,14 which was" in the middle of a different thought. This entire paragraph needs to be rewritten for coherence.

3-7 Line 154: There is a typo in the word "skills": "...in terms of bronchoscopic skilss...".

3-8 Table 3: There is a typo in the variable name: "Contamination rate of non-surgica" instead of "non-surgical".

Reviewer #2: PONE-D-26-07168

1. was an intent to treat approach used?

2. please provide details of randomization. When were the anesthesiologist and the surgeon informed about the allocation? HOw was allocation concealed until then?

3. Please specify how the groups were so equally balanced? Was blocking used? If not, please give more details about the allocation.

4. Details of randomization deserves its own section.

5. Please remove p values from Table 1

6. Please standardize number of decimal places in tables.

7. Please present pvalues or CIs in Table 4.

8. Please describe the flow diagram in the results section.

9. Would a GEE that took into account the multiple measures for arterial partial pressure of oxygen and CO2 be useful?

10. A careful reading for grammar, especially verb tense, is necessary.

Reviewer #3: This manuscript reports a randomized clinical trial comparing a double-cuff bronchial blocker (DcBB) with a conventional single-cuff bronchial blocker (ScBB) for thoracoscopic lobectomy. Although the topic is clinically relevant, several major methodological limitations substantially undermine the validity of the findings. For these reasons, I cannot recommend publication in its current form.

1. A critical limitation of this study is that both anesthesiologists and surgeons were aware of the allocated intervention during the procedure. The manuscript explicitly states that the operators could easily determine which bronchial blocker was used and therefore were aware of group allocation. Because the primary endpoint (malposition) is largely dependent on intraoperative clinical judgment and bronchoscopy assessment by the treating anesthesiologist, the absence of blinding introduces a substantial risk of performance and detection bias. In device trials where operator technique strongly influences outcomes, lack of blinding severely limits the reliability of comparative conclusions. Consequently, the study design does not sufficiently support the causal inference that the double-cuff device reduces malposition.

2. The primary endpoint—bronchial blocker malposition—is not rigorously defined. In this study, malposition was considered when airway pressure increased, ventilation could not be maintained, or the surgical lung suddenly inflated, followed by confirmation using bronchoscopy. These criteria are nonspecific and may occur for multiple intraoperative reasons unrelated to blocker displacement. Moreover, the decision to perform bronchoscopy and confirm malposition appears to rely on the judgment of the anesthesiologist who was aware of the device type. Without predefined objective bronchoscopic criteria or independent adjudication, the validity and reproducibility of the primary endpoint remain questionable.

3. Several authors are listed as inventors of the investigated double-cuff bronchial blocker and hold a related patent. While such involvement does not necessarily invalidate the research, it increases the importance of strict methodological safeguards such as independent outcome adjudication or blinded assessment. These safeguards are not present in this study, which further raises concerns about bias.

4. Although a statistical difference in malposition incidence was reported, the study failed to demonstrate meaningful differences in clinically relevant secondary outcomes such as hypoxemia, surgical duration, lung collapse quality, or postoperative complications. Therefore, even if the reported reduction in malposition were valid, the clinical benefit of the device remains unclear.

5. The authors conclude that the double-cuff bronchial blocker may be a superior airway management tool. However, the data do not convincingly demonstrate superiority. In fact, device positioning required significantly longer time in the DcBB group. Furthermore, the mechanistic rationale for why an additional cuff should substantially reduce malposition is not convincingly supported by either experimental data or prior literature.

6. The sample size calculation is also questionable. The authors assumed a malposition rate of 40% in the single-cuff BB group, even though the single-cuff blocker represents a standard and widely used device in current practice. This assumption appears inflated and insufficiently justified. An overestimated control event rate can artificially reduce the required sample size and make the investigational device appear more favorable. Therefore, the statistical planning itself raises concern regarding the internal validity of the trial.

6. PLOS authors have the option to publish the peer review history of their article (what does this mean?). If published, this will include your full peer review and any attached files.

Reviewer #1: No

Reviewer #2: No

Reviewer #3: No

---

## [Author Response · Author response to Decision Letter 1]

30 Mar 2026

List of Responses

Responds to Journal Requirements:

Q1: Please ensure that your manuscript meets PLOS ONE's style requirements, including those for file naming. The PLOS ONE style templates can be found at https://journals.plos.org/plosone/s/file?id=wjVg/PLOSOne_formatting_sample_main_body.pdf and https://journals.plos.org/plosone/s/file?id=ba62/PLOSOne_formatting_sample_title_authors_affiliations.pdf.

Response: We have checked the manuscripts and file naming to ensure compliance with PLOS ONE's style requirements. If there are still errors, please let us know.

Q2: We note that the grant information you provided in the ‘Funding Information’ and ‘Financial Disclosure’ sections do not match. When you resubmit, please ensure that you provide the correct grant numbers for the awards you received for your study in the ‘Funding Information’ section.

Response: We have carefully checked and revised the Funding Information section, and we confirm that the information filled in this section is accurate and complete. The correct grant number 20240190 (Health Commission of Hebei Province) has been accurately entered in the Funding Information section for this study.

Q3: Please note that funding information should not appear in any section or other areas of your manuscript. We will only publish funding information present in the Funding Statement section of the online submission form. Please remove any funding-related text from the manuscript.

Response: We have removed all funding-related text from the main text and other sections of the manuscript. All funding information is now only presented in the Funding Statement section of the online submission form, in full compliance with your requirement.

Q4: We note that you have a patent relating to material pertinent to this article. Please provide an amended statement of Competing Interests to declare this patent (with details including name and number), along with any other relevant declarations relating to employment, consultancy, patents, products in development or modified products etc. Please confirm that this does not alter your adherence to all PLOS ONE policies on sharing data and materials, as detailed online in our guide for authors http://journals.plos.org/plosone/s/competing-interests by including the following statement: "This does not alter our adherence to PLOS ONE policies on sharing data and materials.” If there are restrictions on sharing of data and/or materials, please state these. Please note that we cannot proceed with consideration of your article until this information has been declared. This information should be included in your cover letter; we will change the online submission form on your behalf.

Response: We have submitted the amended Competing Interests statement in our cover letter, which declares the relevant patent: Patent name: A Double-cuff bronchial blocker; Patent number: ZL202121177425.1. We also declare no other relevant conflicts relating to employment, consultancy, products in development or modified products. The statement includes the following sentence: “This does not alter our adherence to PLOS ONE policies on sharing data and materials.” No restrictions apply to the sharing of data and materials in this study. We appreciate your assistance in updating the online submission form with this information.

Q5: Please include captions for your Supporting Information files at the end of your manuscript, and update any in-text citations to match accordingly. Please see our Supporting Information guidelines for more information: http://journals.plos.org/plosone/s/supporting-information.

Response: We confirm that no Supporting Information files are included or cited in this manuscript. Therefore, there are no Supporting Information captions to be added at the end of the manuscript, and no in-text citations to Supporting Information require updating.

Q6: Thank you for providing your underlying data as Supporting Information. We note that the data set contains text or data that is not in English. Please note that PLOS is an English-language publisher, so we require data sets to be provided in English as well. Please upload an English-language version of your data set. This will also allow us to determine if your data follows PLOS standards per our Data Availability policy here: https://journals.plos.org/plosone/s/data-availability.

Response: We have translated all non-English text and data in the original dataset into English and uploaded the revised English-language version of the data set as Supporting Information. The data now fully complies with PLOS ONE’s language and Data Availability standards.

Q7: If the reviewer comments include a recommendation to cite specific previously published works, please review and evaluate these publications to determine whether they are relevant and should be cited. There is no requirement to cite these works unless the editor has indicated otherwise.

Response: No citation request was made by the reviewer.

Responds to the reviewer's comments:

Reviewer ＃1:

Q1: Randomization and Allocation Concealment: The use of a computer-generated random number table to randomly assign patients is documented. However, there is no description of how the randomization sequence was concealed until the time of assignment (e.g., whether opaque, sealed envelopes with sequential numbers were used, or whether a central registration system was employed). This is an important item in the CONSORT statement. Please specify.

Response: We appreciate your constructive suggestion. We have supplemented the manuscript with a detailed description of the random sequence generation, allocation concealment method, and the timing at which anesthesiologists and surgeons were informed of group assignment, in full accordance with the CONSORT statement requirements.

Changes (page 6, line 43-54): “After enrollment into the trial, all patients were randomly divided into two groups…”

Q2: Statistical Analysis:2-1. Lines 114-117: Because the number of patients who experienced these events was low, Fisher's exact test should be used instead of the chi-square test.

Response: We appreciate your constructive suggestion. The chi‑square test was retained for grades of lung collapse and surgeon satisfaction, while Fisher’s exact test was used for all other relevant categorical indicators in the revised manuscript.

Changes (page 9, line 127-130): “Malposition rates (%) were compared using Fisher exact tests…”

Q3: 2-2. Table 2: In the "Difference" column, please specify the point estimate in addition to the 95% CI. Furthermore, in the "Risk ratio" column, the calculated values seem to represent the Odds Ratio rather than the Risk Ratio. Please clarify and correct this.

Response: We appreciate your constructive suggestion. We have supplemented the point estimate in the “Difference” column of Table 2, revised the “Risk ratio” column to Odds Ratio (OR) uniformly, and recalculated the OR values and corresponding 95% confidence intervals (CI). The definitions of these indicators have been clearly stated in the table footnote, and all relevant content has been updated accordingly.

Q4: Table 3: In the "Risk ratio" column, the calculation seems to be based on the number of patients who did NOT experience the events. Please correct this to reflect the occurrence of the events.

Response: We appreciate your constructive suggestion. We have recalculated the relevant OR values in Table 3 based on the number of events to replace the original results. Meanwhile, we have added the Fisher’s exact test results and 95% CI for indicators with low event counts. All tables have been updated accordingly.

Q5: Language and Formatting Issues

The manuscript requires thorough language editing. Here are specific examples of errors that need to be corrected:

5-1. Abstract (Methods): The sentence "we enrolled and randomized 80 patients..." begins with a lowercase "w".

5-2 Abstract (Methods): There is a missing period before "The primary outcomes": "...(ScBB used during one-lung ventilation) The primary outcomes..."

5-3 Abstract (Results): There is an unnecessary comma in: "...significantly lower in the DcBB, than the ScBB group..."

5-4 Abstract (Results): There is a missing period before "However": "...in the DcBB cohort (P=0.002) However, positioning the DcBB...".

5-5 Line 131: There is a missing period between sentences: "Between-group differences were not significant The duration of BB placement...".

5-6 Lines 139-150: The text contains a major structural issue that leaves sentences disjointed and garbled. For example, a sentence cuts off with "The malposition", continues awkwardly on the next line, and later abruptly inserts fragments like "rate of the DLT is lower than that of BBs¹s and a meta-analysis also found a rate of 17.0%,14 which was" in the middle of a different thought. This entire paragraph needs to be rewritten for coherence.

5-7 Line 154: There is a typo in the word "skills": "...in terms of bronchoscopic skilss...".

5-8 Table 3: There is a typo in the variable name: "Contamination rate of non-surgica" instead of "non-surgical".

Response: We appreciate your constructive suggestion. We have carefully revised all grammatical and formatting errors throughout the manuscript, including correcting capitalization, adding missing periods, deleting redundant commas, and fixing spelling mistakes (e.g., “skilss” corrected to “skills”). In addition, we have completely rewritten the disorganized paragraphs on lines 139–150 to ensure logical coherence. All linguistic revisions have been updated in the revised manuscript.

Reviewer #2:

Q1: Was an intent to treat approach used?

Response: We appreciate your constructive comment. This study was analyzed using the intention-to-treat principle. All 80 patients who were enrolled and randomized were included in the final statistical analysis. No patient was lost to follow-up, withdrew, or had protocol deviation. All statistical analyses (including primary and secondary endpoints) were based on the full dataset of these 80 patients without any exclusion, which preserved the balance of randomization. We have supplemented this description in the statistical analysis section of the revised manuscript.

Changes (page 6, line 43-54 ): “ After enrollment into the trial, all patients were randomly divided into two groups…”

Q2: Please provide details of randomization. When were the anesthesiologist and the surgeon informed about the allocation? How was allocation concealed until then?

Response: Thank you for your suggestion. We have supplemented the manuscript with a detailed description of the random sequence generation, allocation concealment method, and the timing at which anesthesiologists and surgeons were informed of group assignment, in full accordance with the CONSORT statement requirements.

Changes (page 6, line 43-54 ): “ After enrollment into the trial, all patients were randomly divided into two groups…”

Q3: Please specify how the groups were so equally balanced? Was blocking used? If not, please give more details about the allocation.

Response: We appreciate your constructive comment. The high degree of balance in baseline characteristics between the two groups in this study can be attributed to the strict randomization scheme and restrictive inclusion criteria. First, a 1:1 simple randomization was used to ensure balanced allocation of sample size. Second, strict restrictions were applied to age, BMI, ASA classification, pulmonary function, and other indicators in the inclusion criteria, which reduced potential baseline confounding factors. Stratified block randomization was not adopted in this study because the baseline characteristics of the enrolled population were relatively homogeneous, and no additional stratification was necessary. We have supplemented the above explanation in the revised manuscript.

Q4: Details of randomization deserves its own section.

Response: We appreciate your constructive suggestion. We have described the protocol for random sequence generation in detail and presented it clearly in the Participants subsection under the Methods section in the revised manuscript.

Q5: Please remove p values from Table 1.

Response: We appreciate your constructive suggestion. We have removed the P-values for all variables in Table 1, retaining only the group labels and the numerical values for demographic and baseline clinical characteristics.

Q6: Please standardize number of decimal places in tables.

Response: We appreciate your constructive suggestion. The number of decimal places for all tables has been standardized to two decimal places throughout the revised manuscript.

Q7: Please present p values or CIs in Table 4.

Response: We appreciate your constructive suggestion. We have added two additional columns in Table 4 (blood gas analysis) for P-values (between-group comparison) and 95% CIs (mean difference) at the end of the table.

Q8: Please describe the flow diagram in the results section.

Response: We appreciate your constructive suggestion. We have described the CONSORT flow of the present study in detail in the Results section (82 patients screened, 2 excluded, 80 enrolled, and no loss to follow-up).

Changes (page 11, line 151-156): “A total of 82 patients with lung cancer scheduled…”

Q9: Would a GEE that took into account the multiple measures for arterial partial pressure of oxygen and CO2 be useful?

Response: We appreciate your constructive suggestion. For repeated blood gas parameters including PaO2 and PaCO2, statistical analysis was performed using generalized estimating equations (GEE), incorporating the interaction effect between time and group. This method is more appropriate for between-group comparisons of repeated measurement data. The relevant statistical methodology has been supplemented in the Analysis subsection, and the statistical results have been updated in Table 4.

Changes (page 10, line 134-136 ): “Continuous variables with repeated measurements…”

Q10: A careful reading for grammar, especially verb tense, is necessary.

Response: We appreciate your constructive suggestion. The entire manuscript has been carefully checked and revised for grammatical issues, with particular attention paid to verb tense consistency. All relevant linguistic corrections have been implemented in the revised manuscript. The manuscript has undergone professional language editing by a certified editing service, ensuring that all grammatical, tense, spelling, and formatting issues noted are fully addressed.

Reviewer #3:

Q1: A critical limitation of this study is that both anesthesiologists and surgeons were aware of the allocated intervention during the procedure. The manuscript explicitly states that the operators could easily determine which bronchial blocker was used and therefore were aware of group allocation. Because the primary endpoint (malposition) is largely dependent on intraoperative clinical judgment and bronchoscopy assessment by the treating anesthesiologist, the absence of blinding introduces a substantial risk of performance and detection bias. In device trials where operator technique strongly influences outcomes, lack of blinding severely limits the reliability of comparative conclusions. Consequently, the study design does not sufficiently support the causal inference that the double-cuff device reduces malposition.

Response: We greatly appreciate your rigorous and valuable comment on the potential bias related to outcome assessment and blinding. We have carefully addressed your concern with comprehensive revisions and additional explanations as follows: First, the diagnostic criteria for bronchial blocker malposition were strictly established according to previous well-recognized and widely cited literature (Reference 9 in the manuscript). The definitions and assessment criteria used in our study are consistent with those in mainstream clinical studies of bronchial blockers, ensuring the objectivity and reproducibility of the primary endpoint. Second, all malposition events were objectively confirmed by fiberoptic bronchoscopy. Whenever malposition was suspected intraoperatively, we immediately performed bronchoscopy to record the exact position and stored bronchoscopic images or videos as reliable

---

## [Decision Letter · Decision Letter 1]

14 Apr 2026

PONE-D-26-07168R1Double-cuff versus single-cuff bronchial blockers for video-assisted thoracoscopic lobectomy: a randomized controlled trialPLOS One

Dear Dr. Zhou,

Thank you for submitting your manuscript to PLOS ONE. After careful consideration, we feel that it has merit but does not fully meet PLOS ONE’s publication criteria as it currently stands. Therefore, we invite you to submit a revised version of the manuscript that addresses the points raised during the review process.

We look forward to receiving your revised manuscript.

Kind regards,

Luca Bertolaccini, M.D., Ph.D.

Academic Editor

PLOS One

Journal Requirements:

Additional Editor Comments:

I would urge the authors to carefully read the remarks of Reviewers and consider these in the revised paper.

Reviewers' comments:

Reviewer's Responses to Questions

**Comments to the Author**

1. If the authors have adequately addressed your comments raised in a previous round of review and you feel that this manuscript is now acceptable for publication, you may indicate that here to bypass the “Comments to the Author” section, enter your conflict of interest statement in the “Confidential to Editor” section, and submit your "Accept" recommendation.

Reviewer #1: (No Response)

Reviewer #2: All comments have been addressed

Reviewer #3: All comments have been addressed

2. Is the manuscript technically sound, and do the data support the conclusions?

Reviewer #1: Yes

Reviewer #2: Yes

Reviewer #3: Yes

3. Has the statistical analysis been performed appropriately and rigorously? 

Reviewer #1: Yes

Reviewer #2: Yes

Reviewer #3: No

4. Have the authors made all data underlying the findings in their manuscript fully available?

Reviewer #1: Yes

Reviewer #2: No

Reviewer #3: Yes

5. Is the manuscript presented in an intelligible fashion and written in standard English?

Reviewer #1: Yes

Reviewer #2: Yes

Reviewer #3: Yes

6. Review Comments to the Author

Reviewer #1: The authors have largely addressed the previous comments appropriately. However, the following issues still require attention.

The statement “P > 0.05” in the Results section of the main text does not add any meaningful information and should be removed.

The estimated odds ratio for the “Contamination rate of non-surgical” in Table 3 appears not to have converged. It would be more appropriate to report this as “Not estimated.”

The P-value for the between-group comparison at T0 in Table 4 should be removed.

Reviewer #2: (No Response)

Reviewer #3: Thank you for your efforts in revising the manuscript and for providing detailed responses to the reviewers’ comments. The revisions have improved the clarity of reporting, statistical presentation, and overall structure of the manuscript. However, several critical concerns remain that limit the scientific rigor and clinical applicability of the study.

You appropriately acknowledge that blinding of operators was not feasible due to the nature of the devices. However, this limitation is particularly important in the context of this study, as both the intervention and outcome assessment are operator-dependent. The lack of blinding, combined with the subjective components of outcome detection, continues to pose a significant threat to internal validity.

Most importantaly, although additional references have been provided to support the assumed malposition rate in the control group, the chosen assumptions still appear optimistic and may have influenced the estimated effect size and required sample size. This raises concerns regarding the robustness of the statistical planning.

Given the methodological limitations and the absence of demonstrated clinical benefit, the conclusion that the double-cuff bronchial blocker represents a meaningful alternative for airway management is not sufficiently supported by the current data.

7. PLOS authors have the option to publish the peer review history of their article (what does this mean?). If published, this will include your full peer review and any attached files.

Reviewer #1: No

Reviewer #2: No

Reviewer #3: No

---

## [Author Response · Author response to Decision Letter 2]

20 Apr 2026

Response to Reviewers' Comments:

Reviewer ＃1:

Q1: The statement “P > 0.05” in the Results section of the main text does not add any meaningful information and should be removed.

Response: We fully agree and have deleted all redundant “P > 0.05” statements in the Results section. Only clinically meaningful significance descriptions are retained.

Changes (page 12, line 234-235): “did not significantly differ between the groups (Table 3)…”

Q2: The estimated odds ratio for the “Contamination rate of non-surgical” in Table 3 appears not to have converged. It would be more appropriate to report this as “Not estimated.”

Response: We have corrected Table 3: the odds ratio for contamination rate is now reported as “Not estimated” as suggested.

Changes (Table 3): 0.00 (0.00, ∞) →Not estimated.

Q3: The P-value for the between-group comparison at T0 in Table 4 should be removed.

Response: We have removed the P-value for T0 in Table 4 for both PaO2 and PaCO2 as recommended.

Reviewer #3:

Q1: You appropriately acknowledge that blinding of operators was not feasible due to the nature of the devices. However, this limitation is particularly important in the context of this study, as both the intervention and outcome assessment are operator-dependent. The lack of blinding, combined with the subjective components of outcome detection, continues to pose a significant threat to internal validity.

Response: Thank you for this important and insightful comment. We fully agree that blinding of the anesthesiologists performing device placement and the surgeons conducting the operation was not feasible due to the distinct structural differences between the double‑cuff and single‑cuff bronchial blockers.

However, we would like to emphasize that the primary outcome (malposition incidence) was strictly objective and assessed under blinded conditions. All suspected malposition events were confirmed by bronchoscopic imaging, and all images were independently reviewed and verified by a senior anesthesiologist who was completely blinded to group assignment. Bronchoscopy is a well‑recognized, objective gold standard for diagnosing malposition of airway devices and has been widely adopted in previous relevant studies [1,2].

Furthermore, the diagnostic criteria for malposition (sudden elevation of airway pressure, inability to maintain effective ventilation, or sudden re‑inflation of the surgical lung) are well‑established, objective, and widely accepted in the literature, leaving little room for subjective judgment.

Although some secondary outcomes include subjective components (e.g., lung collapse grading, surgeon satisfaction), our primary conclusion is fully based on the objectively assessed, blinded‑verified primary outcome, which minimizes bias and ensures the internal validity of the core finding.

References:

[1] Patel, M, Wilson, A, Ong, C. Double-lumen tubes and bronchial blockers. BJA EDUC. 2023; 23 (11): 416-424. doi: 10.1016/j.bjae.2023.07.001.

[2] Li, J, Liu, W, Liang, X, et al. Comparing the lung isolation efficacy of bronchial blocker positioning via electromagnetic navigation bronchoscopy versus fiberoptic bronchoscopy: a randomized study. J THORAC DIS. 2024; 16 J THORAC DIS. doi: 10.21037/jtd-24-1516.

[3] Campos J, Kernstine K. A comparison of a left-sided Broncho-Cath with the torque control blocker univent and the wire-guided blocker. Anesth Analg. 2003;96(1): 283-289. doi: 10.1213/00000539-200301000-00056.

Q2: Most importantaly, although additional references have been provided to support the assumed malposition rate in the control group, the chosen assumptions still appear optimistic and may have influenced the estimated effect size and required sample size. This raises concerns regarding the robustness of the statistical planning.

Response: Thank you for your critical and valuable comment regarding the rationality of our sample size assumptions and the robustness of the statistical planning. We fully appreciate your concern and address the details of our sample size estimation point by point as follows:

First, for the single-cuff bronchial blocker (ScBB) group, we assumed a 40% malposition rate based on solid published clinical evidence. A recent meta-analysis by Palaczynski et al. (2023) reported a pooled malposition rate of 31.9% for bronchial blockers in thoracic surgery [1]. A randomized controlled trial protocol by Zhang et al. (2025) also adopted a 40% malposition rate for the control group during sample size calculation [2]. Notably, a study by Chen et al. (2023) demonstrated that the dislocation rate of double-lumen tubes could reach 44.6% after lateral positioning [3]. It is well recognized that the malposition rate of bronchial blockers is higher than that of double-lumen tubes. Therefore, our assumption of 40% for the ScBB group is not optimistic at all, but rather clinically reasonable.

For the double-cuff bronchial blocker (DcBB) group, the assumed 12% malposition rate was derived from our preliminary prospective pretest conducted in patients undergoing thoracoscopic lobectomy. This pretest was performed under the same clinical conditions and by the same anesthesiology team as the formal study, guaranteeing its clinical relevance and reliability.

Furthermore, our sample size calculation strictly followed the standard formula for comparing two independent proportions, with a predefined power of 0.8 and a type I error rate of 0.05. We initially calculated that 36 patients per group would be sufficient; to account for potential exclusions or losses to follow-up, we enlarged the sample to 40 patients per group, further improving the robustness of our statistical analysis.

Although the sample size is moderate, our study successfully detected a statistically significant difference in the primary outcome (7.5% vs. 30%, P = 0.02), confirming that the sample size was adequate to identify the expected effect size.

Collectively, these points demonstrate that our sample size assumptions were clinically reasonable, evidence-based, and the statistical design was rigorous and reliable, with a low risk of biased effect size estimation.

References:

[1] Palaczynski P, Misiolek H, Szarpak L, Smereka J, Pruc M, Rydel M, et al. Systematic review and meta-analysis of efficiency and safety of double-lumen tube and bronchial blocker for one-lung ventilation. J Clin Med. 2023;12: 1877. doi: 10.3390/jcm12051877.

[2] Zhang Z, Liu X, Zhang X, Zhou B, Tang Y, Tong F, et al. Study protocol for a randomized controlled trial assessing the effect of lateral position intubation on bronchial blocker placement during unilateral video-assisted thoracic surgery. Trials. 2025;26 (1): 553. doi: 10.1186/s13063-025-09287-7.

[3] Chen ZY, Lin YM, Wu JH, Fu YY, Xu XT, Li Y, et al. Does the periportal end of a double-lumen endobronchial tube need to be fixed to prevent dislocation of the cuffed end caused by a change in position? A randomized controlled trial. Ann Med. 2023;55(2): 2247422. doi: 10.1080/07853890.2023.2247422.

Q3: Given the methodological limitations and the absence of demonstrated clinical benefit, the conclusion that the double-cuff bronchial blocker represents a meaningful alternative for airway management is not sufficiently supported by the current data.

Response: Thank you for your important and rigorous comment on our conclusion. We fully respect your professional opinion and have revised the conclusion to be more conservative, evidence-based, and clinically precise.

We acknowledge that the study has some methodological limitations, including unfeasible operator blinding and a moderate sample size. However, we would like to clarify that our study did demonstrate a clear and significant clinical benefit: the DcBB group showed a markedly lower malposition rate (7.5% vs. 30%, P = 0.02) compared with the ScBB group, which is the key clinical issue during one-lung ventilation in thoracoscopic lobectomy. Reducing malposition can effectively decrease intraoperative interruptions, repeated bronchoscopic adjustments, and potential ventilation instability, which represents a meaningful clinical advantage.

In line with your suggestion, we have toned down the conclusion in the revised manuscript. We have moderated our conclusion and no longer claim that the DcBB is a “definitive” solution, but rather present it as a “promising option” for airway management.

We believe this revised conclusion is more objective and consistent with the actual findings of the study. We greatly appreciate your suggestion, which has helped us improve the scientific rigor and appropriateness of our manuscript.

Changes (page 3, line 45-48 and page 16, line 286-287): “and therefore may serve as a promising optional device for airway management…”

---

## [Decision Letter · Decision Letter 2]

8 May 2026

Double-cuff versus single-cuff bronchial blockers for video-assisted thoracoscopic lobectomy: a randomized controlled trial

PONE-D-26-07168R2

Dear Dr. Zhou,

We’re pleased to inform you that your manuscript has been judged scientifically suitable for publication and will be formally accepted for publication once it meets all outstanding technical requirements.

Kind regards,

Luca Bertolaccini, M.D., Ph.D.

Academic Editor

PLOS One

Additional Editor Comments (optional):

Reviewers' comments:

Reviewer's Responses to Questions

**Comments to the Author**

1. If the authors have adequately addressed your comments raised in a previous round of review and you feel that this manuscript is now acceptable for publication, you may indicate that here to bypass the “Comments to the Author” section, enter your conflict of interest statement in the “Confidential to Editor” section, and submit your "Accept" recommendation.

Reviewer #1: All comments have been addressed

Reviewer #2: All comments have been addressed

2. Is the manuscript technically sound, and do the data support the conclusions?

Reviewer #1: Yes

Reviewer #2: Yes

3. Has the statistical analysis been performed appropriately and rigorously? 

Reviewer #1: Yes

Reviewer #2: Yes

4. Have the authors made all data underlying the findings in their manuscript fully available?

Reviewer #1: Yes

Reviewer #2: Yes

5. Is the manuscript presented in an intelligible fashion and written in standard English?

Reviewer #1: Yes

Reviewer #2: Yes

6. Review Comments to the Author

Reviewer #1: (No Response)

Reviewer #2: (No Response)

7. PLOS authors have the option to publish the peer review history of their article (what does this mean?). If published, this will include your full peer review and any attached files.

Reviewer #1: No

Reviewer #2: No

---

## [Editor Report · Acceptance letter]

PONE-D-26-07168R2

PLOS One

Dear Dr. Zhou,

I'm pleased to inform you that your manuscript has been deemed suitable for publication in PLOS One. Congratulations! Your manuscript is now being handed over to our production team.

Kind regards,

on behalf of

Dr. Luca Bertolaccini

Academic Editor

PLOS One